# An Ecological Study Assessing the Relationship between Public Health Policies and Severity of the COVID-19 Pandemic

**DOI:** 10.3390/healthcare9091221

**Published:** 2021-09-17

**Authors:** Zahra Pasdar, Tiberiu A. Pana, Kai D. Ewers, Weronika A. Szlachetka, Jesus A. Perdomo-Lampignano, David T. Gamble, Sohinee Bhattacharya, Ben Carter, Phyo K. Myint

**Affiliations:** 1Institute of Applied Health Sciences, School of Medicine, Medical Sciences and Nutrition, University of Aberdeen, Aberdeen AB25 2ZD, UK; z.pasdar.17@abdn.ac.uk (Z.P.); tiberiu.pana@abdn.ac.uk (T.A.P.); k.ewers.19@abdn.ac.uk (K.D.E.); weronika.szlachetka.17@abdn.ac.uk (W.A.S.); j.perdomolampignano.15@abdn.ac.uk (J.A.P.-L.); david.gamble@abdn.ac.uk (D.T.G.); sohinee.bhattacharya@abdn.ac.uk (S.B.); 2Department of Biostatistics and Health Informatics, Institute of Psychiatry, Psychology and Neuroscience, King’s College London, London WC2R 2LS, UK; ben.carter@kcl.ac.uk

**Keywords:** COVID-19, public health, ecological study

## Abstract

Reliance on government-led policies have heightened during the COVID-19 pandemic. Further research on the policies associated with outcomes other than mortality rates remains warranted. We aimed to determine associations between government public health policies on the severity of the COVID-19 pandemic. This ecological study including countries reporting ≥25 daily COVID-related deaths until end May 2020, utilised public data on policy indicators described by the Blavatnik school of Government. Associations between policy indicators and severity of the pandemic (mean mortality rate, time to peak, peak deaths per 100,000, cumulative deaths after peak per 100,000 and ratio of mean slope of the descending curve to mean slope of the ascending curve) were measured using Spearman rank-order tests. Analyses were stratified for age, income and region. Among 22 countries, containment policies such as school closures appeared effective in younger populations (r_s_ = −0.620, *p* = 0.042) and debt/contract relief in older populations (r_s_ = −0.743, *p* = 0.009) when assessing peak deaths per 100,000. In European countries, containment policies were generally associated with good outcomes. In non-European countries, school closures were associated with mostly good outcomes (r_s_ = −0.757, *p* = 0.049 for mean mortality rate). In high-income countries, health system policies were generally effective, contrasting to low-income countries. Containment policies may be effective in younger populations or in high-income or European countries. Health system policies have been most effective in high-income countries.

## 1. Introduction

The current COVID-19 global pandemic spreading from the initial outbreak in Wuhan, China, in December 2019 still poses threat to many regions of the world including those that have apparently peaked [1]. As of early September 2021, there have been over 4,638,800 deaths related to COVID-19 globally [2]. Data collected from across the world suggest that the overall case fatality rate is approximately 6%, ranging between 0.55–14.6% across individual countries [3]. Many countries have adopted various lockdown measures and several public health policies to prevent the spread of COVID-19. The United Kingdom, for example, has implemented a strict stay at home policy [4], closing non-essential shops [4], stopping all large social gatherings [4], closing pubs, cafés, restaurants and bars [5] and closing school to almost all children [6], among other measures.

With different types of restrictions being implemented across the world, it is vitally important to assess the effectiveness of the different responses individually and cumulatively. This will allow better understanding of the relationship between these public health policies and relevant outcomes that indicate the severity of the first wave of the pandemic from different dimensions. By doing so, they may be reviewed and revised during current and potential future waves of the COVID-19 pandemic and for future pandemics. To allow comparisons between restrictions imposed by different governments in response to COVID-19, the Blavatnik School of Government of the University of Oxford has developed the Oxford COVID-19 Government Response Tracker (OxCGRT); a platform which collates information across 17 indicators to provide degrees of restriction [7]. They have proposed the Stringency Index (SI) as a tool to allow for day-to-day and between-country comparisons of lockdown measures which aim to reduce civilian activity and social contact [7]. Whether these measures have effect beyond peak spread (and thus deaths) of the pandemic is important to discern in order to help guide future public health policy making.

In this study, we aim to add to the literature by determining the relationship between policy indicators including the stringency index score, and the severity of the first wave of the COVID-19 pandemic using a profiling approach (several outcomes which form the shape of pandemic curve). Thus we not only consider the peak of the first wave of the epidemic curve, but also evolution after the peak, the combination of which provides a better global assessment of the effectiveness of these public health policies.

## 2. Materials and Methods

We used an ecological study design to assess the relationship between policy indicators and their respective stringency index on several outcomes that form the epidemic curve; the mean mortality rate during the rising phase of the curve defined as the mean slope of the mean mortality curve till current peak, time to peak, peak deaths per 100,000 population, cumulative deaths after peak per 100,000 population and the ratio of the mean slope of the descending curve to the mean slope of the ascending curve. We hypothesised that the effectiveness of individual public health policies are unlikely to be “one size fit for all”. Therefore, each analysis was stratified according to age (younger and older populations), income (low- and high-income countries) and region (European and non-European countries), respectively. Due to the ecological study design and the use of publicly accessible data, ethical approval was not required.

### 2.1. Selection of Countries

A total of 22 countries in which the pandemic had reached its peak and which had reported at least 25 daily deaths up till the 31 May 2020 were analysed. These countries included Algeria, Austria, Belgium, Canada, Ecuador, France, Germany, Hungary, Iran, Ireland, Italy, Japan, The Netherlands, Poland, Portugal, Romania, Spain, Sweden, Switzerland, Turkey, UK and the USA.

### 2.2. Definitions of Outcomes, Policy Measures and Stratification Measures

#### 2.2.1. Outcomes

Several parameters of the first wave of the COVID-19 mortality curve were utilised to quantify the severity of the pandemic as well as the evolution of the pandemic after its peak. The severity of the pandemic was quantified using the following measures: (i) the mean mortality rate (ii) time to peak and (iii) the peak number of deaths per 100,000 population. The favourability of the pandemic course after the mortality peaked was quantified using: (i) the cumulative number of deaths recorded after peak, standardised per 100,000 population and (ii) the ratio between the mean slopes of the descending and ascending segments of the mortality curve. The definitions for each of these outcome measures have been listed in Table 1.

All the outcomes were derived from a smoothed mortality curve, obtained by the application of a locally weighted (Lowess) regression using a bandwidth of 0.4 on the raw daily mortality data reported by the World Health Organisation [1]. The peak of the mortality curve was defined as the point at which the first derivate of the Lowess regression line became null. Figure 1 details the derivation of each parameter from the daily mortality data, exemplified using the data from the United Kingdom.

#### 2.2.2. Exposure: Policy Indicators and the Stringency Index

The Oxford COVID-19 Government Response Tacker (OxCGRT) systematically collects data on various public health-related government policies which have been established due to the COVID-19 pandemic [7]. In this case, 17 indicators have been described. These include eight containment and closure indicators (C1: school closures, C2: workplace closures, C3: cancelling of public events, C4: restrictions on gathering size, C5: closing public transport, C6: stay at home requirements, C7: restrictions on internal movement, C8: restrictions on international travel), four economic response indicators (E1: income support, E2: debt contract/relief for households, E3: fiscal measures, E4: giving international support) and five health systems indicators (H1: public information campaigns, H2: testing policy, H3: contact tracing, H4: emergency investment in healthcare, H5: investment in COVID-19 vaccines) [8]. All indicators are measured using a simple ordinal scale except for five (E3, E4, H4, H5 and M1). Indicators E3, E4, H4 and H5 were numeric and numeric indicators were typically measured in the value United States Dollar (USD) and M1 represents a miscellaneous indicator which included free text data [8]. Indicators (e.g., stay at home requirements) are assigned a score (e.g., 0–3) based upon the strictness of each policy. A total stringency index has been calculated using only policy indicators (C1–C8 and H1). Details of the approach to scoring of indicators and formulae for calculation of the stringency index has been previously described [9].

For the analyses evaluating the mean mortality rate, time to peak and the number of peak deaths per 100,000 population, the mean SI and its indicators were calculated for each of the 22 countries from the first day when more than 2 COVID-19 deaths were reported until two weeks before the peak of the mortality curve. For the analyses evaluating the cumulative number of deaths after the pandemic peak as well as the slope of the descending mortality curve, the mean SI and its indicators were calculated for each country from the first day when more than 2 COVID-19 deaths were reported until the 17 May 2020 (2 weeks before the end of the study period). A two-week delay between the exposure and the measured outcomes from starting point to end point was implemented to allow for these restrictions to have an effect.

Data used for the stratification of analyses were collected from publicly accessible resources. The median population age was extracted from the United Nations World Population Prospects [10]. Country income data were extracted as GDP per capita by Purchasing Power Parity (PPP) in current international dollars in the year 2018, from the World Bank [11].

### 2.3. Statistical Analysis

All analyses were performed in Stata 15.1SE, Stata Statistical Software. A 5% threshold of statistical significance was utilised for all analyses (*p* < 0.05). Spearman rank-order correlation coefficients were computed to measure the strength of association between eligible policy indicators and SI against each of the five outcomes. Results were stratified by the median value of the median country age (42.133) into those with younger (Belgium, Canada, Algeria, Ecuador, the United Kingdom, Ireland, Iran, Poland, Sweden, Turkey and the United States) and older populations (Austria, Switzerland, Germany, Spain, France, Hungary, Italy, Japan, the Netherlands, Portugal and Romania). Stratification was also performed by median country GDP per capita as cut off point ($45,342) into low- (Algeria, Ecuador, Spain, France, Hungary, Iran, Italy, Japan, Poland, Portugal, Romania and Turkey) and high-income countries (Austria, Belgium, Canada, Switzerland, Germany, the United Kingdom, Ireland, the Netherlands, Sweden and the United States) as well as European (Austria, Belgium, Switzerland, Germany, Spain, France, the United Kingdom, Hungary, Ireland, Italy, The Netherlands, Poland, Portugal, Romania and Switzerland) and non-European countries (Canada, Algeria, Ecuador, Iran, Japan, Turkey, the United States). All outcomes were transformed using a natural logarithm prior to analysis.

## 3. Results

A total of 22 countries during the first wave in which the pandemic had reached its peak, defined as constant decline in mortality since then, and which had at least 25 daily deaths reported up till the 31 May 2020, were included. Appendix A demonstrates the data used in this study.

### 3.1. Policy Indicators and Median Age

Table 2 details Spearman correlation coefficients from different policy indicators and the stringency index against each of the five outcome measures, stratified by younger population age group, based on median country age. Figure 2 details the key for visual interpretation of Table 2 and all subsequent tables. Containment and closure indicators were mostly associated with good outcomes. Notably, the cancellation of public events had the strongest negative association between peak deaths per 100,000 (r_s_ = −0.800, *p*-value = 0.003) and cumulative deaths after peak, per 100,000 (r_s_ = −0.790, *p*-value = 0.004). Closure of schools was effective against all outcomes, except time to peak, and the strongest association was seen in cumulative deaths after peak, per 100,000 (r_s_ = −0.744, *p*-value = 0.009). Economic response indicators and health system indicators were generally associated with bad outcomes for number of deaths and mean mortality rates, however they were associated with a more controlled pandemic peak and decline. Contact tracing had the strongest positive association with peak deaths per 100,000 and this result was statistically significant (r_s_ = 0.790, *p*-value = 0.004). All 17 indicators had a positive association with speed of decline of the number of daily reported deaths over the rate at which the mortality curve peaked, except fiscal measures.

**Table 2 healthcare-09-01221-t002:** Spearman rank-order correlation coefficient of policy indicators and the Stringency Index against five different measures of pandemic severity in the young population based on median country age.

Index	Description	Log Mean Mortality Rate	Log Time to Peak (Days)	Log Peak Deaths/100,000 Pop.	Log Cumulative Deaths after Peak/100,000 Pop.	Log Slope of Descending Curve/Slope of Ascending Curve
C1	School Closing	−0.600	−0.306	−0.620 *	−0.744 *	0.297
C2	Workplace Closing	−0.279	0.000	−0.018	−0.009	0.845 *
C3	Cancel Public Events	−0.569	−0.351	−0.800 *	−0.790 *	0.200
C4	Gathering Restrictions	−0.211	0.241	0.349	0.118	0.391
C5	Close Public Transport	−0.340	−0.295	−0.317	−0.391	0.317
C6	Stay At Home Requirement	−0.360	−0.385	−0.342	−0.273	0.409
C7	Restriction on Internal Movement	−0.400	−0.051	−0.155	−0.227	0.627 *
C8	International Travel Controls	−0.439	0.153	−0.403	−0.352	0.334
E1	Income Support	−0.014	0.172	0.718 *	0.633 *	0.256
E2	Debt/Contract Relief	−0.060	0.130	0.501	0.431	0.514
E3	Fiscal Measures	0.395	0.399	0.578	0.464	−0.073
E4	International Support	0.630 *	0.547	0.341	0.390	0.305
H1	Public Information Campaigns	−0.156	0.073	0.439	0.364	0.376
H2	Testing Policy	0.258	0.674 *	0.268	0.270	0.642 *
H3	Contact Tracing	0.270	0.399	0.790 *	0.480	0.308
H4	Emergency Investment in Healthcare	0.498	0.242	0.395	0.633 *	0.477
H5	Investment in Vaccines	0.630 *	0.547	0.341	0.560	0.431
SI	Stringency Index	−0.536	−0.184	−0.236	−0.382	0.518

* Indicates statistical significance; *p*-value < 0.05. The key to visual interpretation for the table has been described in Figure 2. A more saturated green represents a stronger association to a good outcome, and darker red represents a stronger association to a bad outcome.

**Figure 2 healthcare-09-01221-f002:**
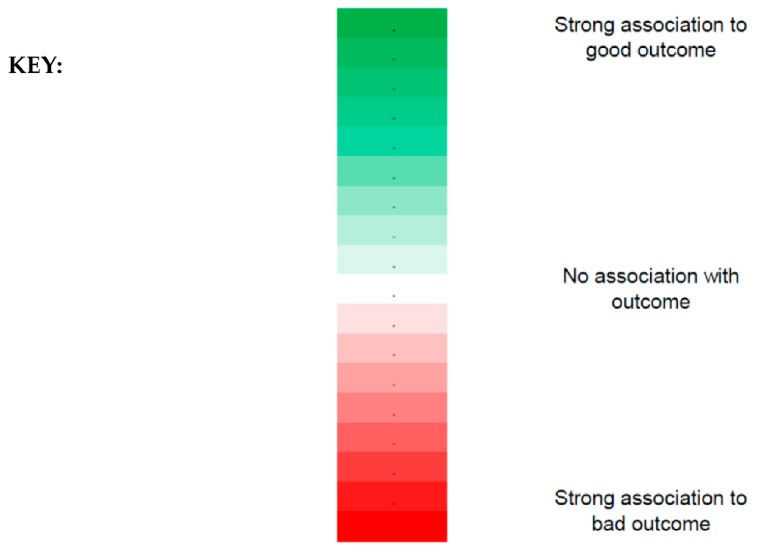
Key to visual interpretation of Table 2, Table 3, Table 4, Table 5, Table 6 and Table 7. The strength of association can be visualised using the colour gradient denoted above where a more saturated green represents a stronger association to a good outcome, and darker red represents a stronger association to a bad outcome. In contrast, the lighter the shades of red or green, the closer the representation to having no association with the outcome (white).

Table 3 details correlation coefficients of policies across the five outcomes in older populations based on median country age. In older populations, debt or contract relief had the strongest association with good outcomes: a significant negative association with mean mortality rate (r_s_ = −0.725, *p*-value = 0.046), peak deaths per 100,000 (r_s_ = −0.743, *p*-value = 0.009) and cumulative deaths after peak, per 100,000 (r_s_ = −0.761, *p*-value = 0.007). Workplace closures were not associated with lower cumulative deaths after peak, per 100,000 (r_s_ = 0.752, *p*-value = 0.008) and investment in vaccines was associated with higher mean mortality rate (r_s_ = 0.665, *p*-value = 0.052).

**Table 3 healthcare-09-01221-t003:** Spearman rank-order correlation coefficient of policy indicators and the Stringency Index against five different measures of pandemic severity in the old population based on median country age.

Index	Description	Log Mean Mortality Rate	Log Time to Peak (Days)	Log Peak Deaths/100,000 Pop.	Log Cumulative Deaths after Peak/100,000 Pop.	Log Slope of Descending Curve/Slope of Ascending Curve
C1	School Closing	−0.216	0.095	−0.081	0.186	0.577
C2	Workplace Closing	−0.079	−0.471	0.126	0.752 *	−0.606 *
C3	Cancel Public Events	0.054	−0.576	0.202	0.202	−0.607 *
C4	Gathering Restrictions	0.005	−0.164	0.023	0.145	−0.136
C5	Close Public Transport	−0.315	−0.054	−0.282	0.005	−0.041
C6	Stay At Home Requirement	−0.156	−0.203	−0.092	0.373	0.018
C7	Restriction on Internal Movement	−0.075	0.225	−0.084	−0.101	0.330
C8	International Travel Controls	−0.047	0.159	−0.075	−0.064	0.266
E1	Income Support	0.114	−0.206	−0.023	−0.064	−0.187
E2	Debt/Contract Relief	−0.725 *	0.046	−0.743 *	−0.761 *	−0.109
E3	Fiscal Measures	0.229	0.092	0.211	0.255	0.519
E4	International Support	−0.400	0.502	−0.500	−0.232	0.569
H1	Public Information Campaigns					
H2	Testing Policy	−0.094	0.154	−0.129	−0.033	−0.060
H3	Contact Tracing	0.119	0.065	0.278	0.196	−0.084
H4	Emergency Investment in Healthcare	0.208	−0.148	0.162	0.073	0.624 *
H5	Investment in Vaccines	0.665 *	0.052	0.451	0.268	0.045
SI	Stringency Index	−0.355	−0.219	−0.218	0.400	−0.073

* Indicates statistical significance; *p*-value < 0.05. The key to visual interpretation for the table has been described in Figure 2. A more saturated green represents a stronger association to a good outcome, and darker red represents a stronger association to a bad outcome.

### 3.2. Policy Indicators and Income

Table 4 and Table 5 detail the associations between policy indicators and stringency index against outcomes in low- and high-income countries, based on median GDP per capita (PPP), respectively. In low-income countries (Table 4), workplace closures were not effective against cumulative deaths after peak, per 100,000 (r_s_ = 0.698, *p*-value = 0.012). Similarly, investment in vaccines was associated with higher mean mortality rate (r_s_ = 0.640, *p*-value = 0.025). The strongest association with a good outcome was between public information campaigns and the speed of decline of number of daily reported deaths over the rate at which the mortality curve peaked (r_s_ = 0.640, *p*-value = 0.025).

Among high-income countries (Table 5), containment strategies were mostly effective, and the strongest association was seen between restrictions on internal movement and ratio of slope of descending curve over slope of ascending curve (r_s_ = 0.915, *p*-value < 0.001). Emergency investment in healthcare had the strongest positive correlation with mean mortality rate (r_s_ = 0.825, *p*-value = 0.003).

**Table 4 healthcare-09-01221-t004:** Spearman rank-order correlation coefficient of policy indicators and the Stringency Index against five different measures of pandemic severity in low-income countries based on median GDP per capita (PPP).

Index	Description	Log Mean Mortality Rate	Log Time to Peak (Days)	Log Peak Deaths/100,000 Pop.	Log Cumulative Deaths after Peak/100,000 Pop.	Log Slope of Descending Curve/Slope of Ascending Curve
C1	School Closing	−0.218	−0.044	0.131	−0.221	0.379
C2	Workplace Closing	−0.014	0.043	0.459	0.698 *	0.061
C3	Cancel Public Events	0.048	−0.503	0.177	0.177	−0.462
C4	Gathering Restrictions	0.090	0.202	0.463	0.245	−0.158
C5	Close Public Transport	−0.140	−0.084	0.119	−0.028	−0.063
C6	Stay At Home Requirement	−0.063	−0.250	0.162	0.168	−0.042
C7	Restriction on Internal Movement	−0.145	0.323	0.078	−0.196	−0.035
C8	International Travel Controls	−0.276	0.014	0.108	0.000	0.388
E1	Income Support	0.198	0.148	0.321	0.305	0.298
E2	Debt/Contract Relief	−0.455	0.336	0.011	0.032	0.361
E3	Fiscal Measures	0.036	0.200	0.174	0.225	0.232
E4	International Support	−0.393	0.481	−0.480	−0.177	0.462
H1	Public Information Campaigns	−0.086	0.291	0.371	0.231	0.640 *
H2	Testing Policy	0.053	0.497	0.374	0.366	0.535
H3	Contact Tracing	0.106	0.368	0.531	0.477	0.466
H4	Emergency Investment in Healthcare	−0.004	−0.446	0.204	0.225	0.421
H5	Investment in Vaccines	0.640 *	−0.127	0.570	0.542	0.028
SI	Stringency Index	−0.105	0.210	0.308	0.301	0.168

* Indicates statistical significance; *p*-value < 0.05. The key to visual interpretation for the table has been described in Figure 2. A more saturated green represents a stronger association to a good outcome, and darker red represents a stronger association to a bad outcome.

**Table 5 healthcare-09-01221-t005:** Spearman rank-order correlation coefficient of policy indicators and the Stringency Index against five different measures of pandemic severity in high-income countries based on median GDP per capita (PPP).

Index	Description	Log Mean Mortality Rate	Log Time to Peak (Days)	Log Peak Deaths/100,000 Pop.	Log Cumulative Deaths after Peak/100,000 Pop.	Log Slope of Descending Curve/Slope of Ascending Curve
C1	School Closing	−0.638 *	−0.492	−0.725 *	−0.480	0.498
C2	Workplace Closing	−0.460	−0.468	−0.288	−0.091	0.503
C3	Cancel Public Events	−0.526	−0.606	−0.683 *	−0.731 *	0.116
C4	Gathering Restrictions	−0.723 *	−0.543	−0.717 *	−0.612	0.552
C5	Close Public Transport	−0.116	−0.154	−0.382	−0.123	0.175
C6	Stay At Home Requirement	−0.491	−0.766 *	−0.576	0.358	0.636 *
C7	Restriction on Internal Movement	−0.207	−0.171	−0.650 *	−0.030	0.915 *
C8	International Travel Controls	−0.117	0.135	−0.534	−0.448	0.190
E1	Income Support	−0.446	−0.039	−0.123	−0.162	0.019
E2	Debt/Contract Relief	−0.227	−0.283	−0.166	−0.117	0.337
E3	Fiscal Measures	0.697 *	0.292	0.358	0.370	0.115
E4	International Support	0.634 *	0.711 *	−0.007	−0.319	0.325
H1	Public Information Campaigns	−0.522	−0.466	0.058	−0.058	−0.290
H2	Testing Policy	0.022	0.389	−0.529	−0.485	0.265
H3	Contact Tracing	−0.068	0.110	0.143	−0.330	−0.381
H4	Emergency Investment in Healthcare	0.825 *	0.596	−0.063	0.200	0.479
H5	Investment in Vaccines	0.731 *	0.630	−0.191	−0.178	0.595
SI	Stringency Index	−0.552	−0.462	−0.479	−0.139	0.370

* Indicates statistical significance; *p*-value < 0.05. The key to visual interpretation for the table has been described in Figure 2. A more saturated green represents a stronger association to a good outcome, and darker red represents a stronger association to a bad outcome.

### 3.3. Policy Indicators and Geographical Region

Table 6 details the correlation coefficients between policies and outcomes in European countries and Table 7, in non-European countries, respectively. In European countries (Table 6) most indicators except cancellation of public events and contact tracing were positively associated with the ratio of the slope of the descending curve over the ascending curve. Of these, restrictions on internal movements had the strongest association and this result was statistically significant (r_s_ = 0.640, *p*-value = 0.010). Investment in vaccines was most strongly associated with a bad outcome in terms of the mean mortality rate (r_s_ = 0.674, *p*-value = 0.006). Additionally, the measured stringency index for European countries was strongly associated with mean mortality rate and peak deaths per 100,000 population, and these results were statistically significant. In non-European countries (Table 7), school closures were most effective against mean mortality rate (r_s_ = −0.757, *p*-value = 0.049) and testing policy had the strongest positive association with peak deaths per 100,000 (r_s_ = 0.847, *p*-value = 0.016).

**Table 6 healthcare-09-01221-t006:** Spearman rank-order correlation coefficient of policy indicators and the Stringency Index against five different measures of pandemic severity in European countries.

Index	Description	Log Mean Mortality Rate	Log Time to Peak (Days)	Log Peak Deaths/100,000 Pop.	Log Cumulative Deaths after Peak/100,000 Pop.	Log Slope of Descending Curve/Slope of Ascending Curve
C1	School Closing	−0.387	−0.289	−0.523 *	−0.287	0.306
C2	Workplace Closing	−0.307	−0.433	−0.197	0.287	0.161
C3	Cancel Public Events	−0.463	−0.535 *	−0.610 *	−0.600 *	−0.055
C4	Gathering Restrictions	−0.320	−0.067	−0.333	−0.218	0.363
C5	Close Public Transport	−0.372	−0.048	−0.445	−0.229	0.332
C6	Stay At Home Requirement	−0.448	−0.202	−0.452	0.168	0.568 *
C7	Restriction on Internal Movement	−0.198	0.098	−0.271	−0.145	0.640 *
C8	International Travel Controls	−0.398	0.096	−0.562 *	−0.536 *	0.167
E1	Income Support	−0.096	0.133	0.100	0.067	0.202
E2	Debt/Contract Relief	−0.440	−0.090	−0.349	−0.238	0.315
E3	Fiscal Measures	0.563 *	−0.074	0.358	0.425	0.200
E4	International Support	0.371	0.187	0.247	0.085	0.434
H1	Public Information Campaigns					
H2	Testing Policy	−0.262	0.238	−0.319	−0.151	0.181
H3	Contact Tracing	0.096	0.188	0.338	0.141	−0.063
H4	Emergency Investment in Healthcare	0.383	−0.065	0.274	0.496	0.429
H5	Investment in Vaccines	0.674 *	0.171	0.321	0.358	0.497
SI	Stringency Index	−0.629 *	−0.189	−0.604 *	−0.143	0.414

* Indicates statistical significance; *p*-value < 0.05. The key to visual interpretation for the table has been described in Figure 2. A more saturated green represents a stronger association to a good outcome, and darker red represents a stronger association to a bad outcome.

**Table 7 healthcare-09-01221-t007:** Spearman rank-order correlation coefficient of policy indicators and the Stringency Index against five different measures of pandemic severity in non-European countries.

Index	Description	Log Mean Mortality Rate	Log Time to Peak (Days)	Log Peak Deaths/100,000 Pop.	Log Cumulative Deaths after Peak/100,000 Pop.	Log Slope of Descending Curve/Slope of Ascending Curve
C1	School Closing	−0.757 *	0.046	−0.223	−0.401	0.223
C2	Workplace Closing	0.000	0.111	0.643	0.750	0.321
C3	Cancel Public Events	0.134	−0.693	0.134	0.134	−0.490
C4	Gathering Restrictions	0.158	0.429	0.788 *	0.432	−0.198
C5	Close Public Transport	−0.148	−0.423	0.111	0.108	−0.144
C6	Stay At Home Requirement	−0.250	−0.482	0.107	0.071	−0.357
C7	Restriction on Internal Movement	0.000	−0.185	0.536	0.214	−0.107
C8	International Travel Controls	−0.090	0.037	0.559	0.450	0.396
E1	Income Support	0.144	0.486	0.775 *	0.536	0.000
E2	Debt/Contract Relief	−0.306	0.879 *	0.180	−0.036	0.536
E3	Fiscal Measures	0.037	0.846 *	0.408	0.286	0.393
E4	International Support	−0.118	0.797 *	0.079	−0.020	0.493
H1	Public Information Campaigns	−0.374	0.511	0.158	−0.020	0.670
H2	Testing Policy	0.288	0.486	0.847 *	0.714	0.571
H3	Contact Tracing	0.000	0.656	0.474	0.177	0.512
H4	Emergency Investment in Healthcare	0.148	−0.077	0.519	0.342	0.739
H5	Investment in Vaccines	0.535	0.324	0.802 *	0.668	0.178
SI	Stringency Index	−0.143	−0.148	0.393	0.250	−0.036

* Indicates statistical significance; *p*-value < 0.05. The key to visual interpretation for the table has been described in Figure 2. A more saturated green represents a stronger association to a good outcome, and darker red represents a stronger association to a bad outcome.

## 4. Discussion

In this study evaluating the association between government-led public health policies and the severity of the first wave of the COVID-19 pandemic using profiling approach, we found that containment and closure policies were generally effective in younger populations and high-income countries, and debt/contract relief in older populations. Similarly, containment and closure policies were generally associated with good outcomes in European countries, whereas in non-European countries, school closures alone had the most favourable association with several outcomes. Moreover, health system policies did not appear to be associated with better outcomes in low-income countries, in contrast to high-income countries, where policies for testing were generally effective, along with closure and containment measures. To the best of our knowledge, this is one of few studies evaluating the association of different government policies on a number of different outcome measures that form the components of the first wave of the COVID-19 pandemic severity.

In a recent nationwide preprint study assessing the impact of lockdown measures on COVID-19 mortality and case numbers, the authors found that early introduction for every government policy with the exception of testing policy, contact tracing and workplace closures, was associated with reduced mortality and case numbers [12]. They noted that the size of effect of introducing such measures at an early stage, such as school closure policies being implemented 24 days earlier, was associated with halving of the mortality as of the 29 April 2020 [12]. Similarly, our analysis showed limited benefit of contact tracing and testing policy, though some benefit was seen in high-income countries. This finding may reflect that high-income countries are better equipped with specialised diagnostic facilities and have access to formal healthcare systems [13]. Yet, despite this apparent advantage, it should also be appreciated that the formation of a successful contact tracing programme is likely more complex than solely having access to high-quality resources, and many other factors may come into play, for example, the ability to install and maintain adequate organisational and leadership approaches. This may explain the observation that, indeed not all high-income countries were in fact successful with regards to their contact tracing [14].

Moreover, a study with data compiled from 1717 local, regional and national non-pharmaceutical interventions deployed across China, South Korea, Italy, Iran, France and the United States (US) found that anti-contagion policies have significantly slowed the growth rate of COVID-19 infections [15]. They estimated that in the absence of such policy actions, early infections of COVID-19 exhibit exponential growth rates of roughly 38% per day [15].

Undoubtedly, there is evidence to support the beneficial impact of such policies on public health [12,15,16,17,18] and the timing at which these policies are introduced are important [12,15]. However, some policies can be more effective than others, and tailoring these measures with knowledge of population characteristics may allow for more tactful intervention.

In younger populations, we observed that containment and closure policies, in particular school closures and cancellation of public events, had the most statistically significant associations with good outcomes, rather than economic and health policies. A systematic review on school closure and management practices during coronavirus outbreaks reported limited and inconsistent results in the literature on the effectiveness of school closures during previous coronavirus outbreaks, such as SARS [19]. However, they reported that recent modelling studies of COVID-19 predict that school closures alone would prevent 2–4% of deaths [19]. Nevertheless, the role of children in the transmission of COVID-19 is still to be elucidated [20] and in contrast with Influenza where children appear to be a key source of transmission, children are more likely to have milder or asymptomatic forms of COVID-19 and are less likely to transmit the disease whilst coughing or sneezing, despite having comparable infection rates to adults [19].

A strong association was found with regards to cancellation of public events and lower peak deaths per 100,000 in the stratification of countries with a younger median age, and this may mirror the fact that younger age groups are more likely to be involved in public gatherings and mass events. A preprint study from Japan reported that to control COVID-19 outbreak, voluntary event cancellations took place from 26 of February to 11 of March where sports and entertainment events were cancelled [21]. The authors found that such measures can reduce COVID-19 infectiousness by 35% but the reproduction number remains higher than one [21].

We also found that contact tracing was positively associated with peak deaths per 100,000 in countries with a younger median age population. It could be plausible that this finding may be attributable to reverse causation, where contact tracing measures have subsequently heightened following government alerting of increasing COVID-19 deaths.

Among health system policies, we found that investment in vaccines was positively associated with mean mortality rates in countries stratified according to older median age. This policy relates to announced public spending on vaccine development [8]. Given that there was no available vaccine during this time, the expenditure in this domain by governments may still necessitate time to allow its benefits in the long-run to be observed. Additionally, in non-European countries, we observed that testing policy had the strongest positive association with peak deaths per 100,000. The testing policy indicator describes who can be tested in a country. Thus, in countries where testing was extensively performed, such as publicly available testing, a higher number of COVID-19 related deaths may have consequently been recorded.

### Strengths and Limitations

The main strength of this study is the use of various outcome measures to assess the severity of the pandemic and its evolution after peak, which provides a broader picture of pandemic severity (using a profiling approach), compared to solely using mere mortality peak/rates. Whilst outcome measures such as death rates and case fatality rates may be difficult to compare because of differences in testing rates across countries and true counts of actual deaths, we utilised the mean slope of the mortality curve amongst other parameters which allows for more comparable measures.

This study has some limitations. Whilst daily mortality data were extracted from a reputable source (World Health Organization), we appreciate that deaths may be underreported, particularly if occurring out with hospitals. Whilst we have utilised all policy indictors recorded by the Blavatnik school of Government to be considered in our analysis, data on important public health measures such as the use of personal protective equipment (PPE) including usage of face masks, were not available. However, due to the timing of our study which covered the early phase of the pandemic, countries such as the UK still followed strict stay at home requirements which could make measuring the true effects of PPE usage in public spaces more difficult to determine. Additionally, government policies may be placed into effect in rapid succession and thus true independent associations between a policy measure and outcome may be difficult to disentangle. It is also important to acknowledge that much of public health and disease prevention occurs at the local level [15,22]. Therefore, the effect of local and regional policies on managing the COVID-19 pandemic in different countries may be overlooked, given the national scope of the data used in the analysis. It is therefore likely that greater inclusion and awareness of local health policies would play a major role in mitigating spread of COVID-19. By setting eligibility criteria to only include countries with at least 25 COVID-related deaths our sample size was modest. Further, this research followed an ecological study design and the limitations and biases conforming to the nature of such study designs must be acknowledged. We cannot exclude the ecological fallacy nor reverse causation, and it is important to note that at best, our results are only hypothesis generating. Moreover, it must be acknowledged that significant variance may exist within the stratification groups for age and income groups.

## 5. Conclusions

There is evidence of the benefits of public health policies on mitigating the severity of the first wave of the COVID-19 pandemic, and these benefits differ according to age of population, country-level incomes and world region. Different policies also have different impact on the different phases of the pandemic. Moreover, it is important to acknowledge that the implementation of policies can often be accompanied with potential detrimental economic and social effects, and therefore operation of these measures may need to be undertaken strategically and dynamically. Future research using longitudinal data on the implementation of public health policies and successive health outcomes from various COVID-19 affected countries is warranted.

## Figures and Tables

**Figure 1 healthcare-09-01221-f001:**
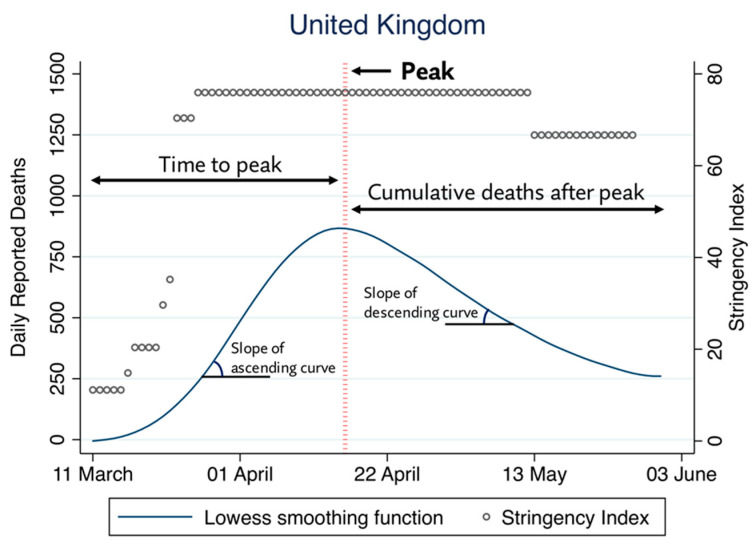
Derivation of parameters for outcome measures from daily mortality data, using data from the United Kingdom as an example.

**Table 1 healthcare-09-01221-t001:** Outcome measures and their definitions.

Aspect of Pandemic Assessed	Outcome Measure	Outcome Measure Definition
Severity of pandemic	Mean mortality rate	The mean slope of the mean mortality curve in each country during the ascending phase of the mortality curve, defined from the first day when more than 2 COVID-19 deaths were reported until the mortality curve reached its peak value.
	Time to peak	Defined as the number of days from the first day when more than 2 COVID-19 deaths were reported in each country until the mortality curve reached its current peak value.
	Peak number of deaths per 100,000 population	The peak number of deaths per 100,000 population.
Favourability of the pandemic course after the mortality peaked	Cumulative number of deaths recorded after peak, per 100,000 population	Cumulative number of deaths recorded after peak until the 31 May 2020, standardised per 100,000 population.
	Ratio between the mean slopes of the descending and ascending segments of the mortality curve	The ratio between the mean slopes of the descending and ascending segments of the mortality curve, as a quantification of the speed of the decline of the number of daily reported deaths adjusted for the rate at which the mortality curve peaked.

## Data Availability

The data presented in this study are available in the online Appendix A.

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
