# Peer review of "An Ecological Study Assessing the Relationship between Public Health Policies and Severity of the COVID-19 Pandemic"

_healthcare, 2021, doi:10.3390/healthcare9091221_

Round 1

Reviewer 1 Report

This is a comprehensive report on the descriptive analysis of the relationship between public policies and severities of pandemic such as COVID-19 in selected countries.  The paper is readable and very informative about the relevance of public health policies to the severity of COVID-19.  Two areas could be strengthened: 1) the diversity of countries selected is a concern since only aggregated or ecological variables are used in the analysis; and 2) the confusion of tables presented.  My suggestions are: 1)  to fully develop a future research agenda using longitudinal data available from WHO or OECD countries; and 2) to clarify the gradients of statistical significance (using color to differentiate the level of significance).  Perhaps, in each table you could add a note to reflect the meanings of each color gradient.

In conclusion, I think that the paper is well formulated and presented. Minor amendments are recommended as noted above.

Author Response

Thank you very much for your helpful and insightful feedback on our manuscript. Whilst the diversity of countries selected for the analysis is limited compared to the overall number of countries affected by the COVID-19 pandemic, this is due to our pre-specified inclusion criteria and the time frame during which the analyses were conducted. To have sufficient sample size for the outcomes of our analysis in order to draw meaningful associations, we included countries in which the pandemic had reached its peak and which had reported at least 25 daily deaths up till the 31st of May, 2020. Therefore, only 22 countries fulfilled these criteria from the data, as stated in the Materials and methods section of our paper (lines 76-80, page 2). However, we have included your important suggestion for future research under conclusions, lines 340-342, page 15:

“Future research using longitudinal data on the implementation of public health policies and successive health outcomes from various COVID-19 affected countries is warranted.”

Thank you for your comments on the presentation of our tables. We have used colour to visually signify the strength of association towards a good or bad outcome from our analyses, and as you have suggested, we have included further details of its visual interpretation in the footnote of figure 2, lines 192-196, page 6:

Figure 2. Key to visual interpretation of tables 1-6. The strength of association can be visualised using the colour gradient denoted above where a darker green represents a stronger association to a good outcome, and darker red represents a stronger association to a bad outcome. In contrast, the lighter the shades of red or green, the closer the representation to having no association with the outcome.” 

Additionally, we have used an asterisk to denote which of these values reached the cut-off value regarded as being statistically significant (p-value <0.05). Since some tables had only a few results reaching the cut-off value for statistical significance, we believed using a colour gradient to signify strength of associations for the results would be more meaningful to compare the different policy indicators.

Reviewer 2 Report

This is an interesting paper that examines 22 countries in relation to the policies that they have implemented and the impact on the severity of the pandemic using a set of indicators designed by the Blavatnik School Government. While the analysis is interesting and reveals some interesting correlations I feel there are very significant problems with the way the indicators have been operationalised as it aggregates very different countries and is insufficiently sensitive to variation in practice.

The countries in the study are grouped, and presented as binomials (to facilitate the logistic regression), and this hides significant variation in wealth and age of the population. Indeed, the use of GDP (PPP) takes no account of income disparity that may be a far better indicator of the size of deprived population, or government income/budget per capita which may be a better indicator of the capacity of government to take action.

While an attempt has been made to consider differences in population demography is this very simplistic – again a bivariate variable based on age – but this is also a relative judgement. There is no attempt, for instance to look at the proportion of older people (the most vulnerable to COVID-19) in a population which could have been done with a different variable, or any attempt to consider issues of gender or ethnicity (both of which have implications for both infection and hospitalisation rates). This approach leads to statements that appear misleading by referring to countries, for instance, as ‘low income’ or ‘young’ when these are relative judgements and incorporate significant variation.

In terms of the policy side the variables are also very insensitive. Twelve policies are scored (1-3) based on perceived ‘stringency’; there is no indication of how the strictness of a policy, the scoring, is arrived at. On line 119, five indicators are identified that are not scored in this way, but there is no explanation of how their values are arrived at.

The Stringency Index is included in the analysis and reported in the tables but is not commented upon. Indeed I am not sure that the Stringency Index adds anything to the analysis or article.

There is no policy indicator related to required mask use in public spaces which is a significant gap. Secondly, policies were not all implemented simultaneously but evolved so using this categorical approach may not be appropriate; there is no comment on this.

None of these really significant limitations are considered in the Strengths and Limitations section (p.15).

The consequence is that while conclusions can be drawn from the analysis I am not sure how they can be interpreted as the definition of the variables is so aggregated. Personally, I have little faith in the conclusions that are drawn despite their alignment with other research. Given the insensitivity of the analysis I am not sure what this adds.

For this article to be valuable it needs to approach the analysis in a far more sophisticated way rather than glossing over the significant variation between countries in terms of policies and implementation.

Reviewer 3 Report

The manuscript provides an assessment of the associations between various public health interventions utilized at national levels to manage the infection and mortality risks associated with COVID-19 during the first wave of the pandemic in the nations included within the study.

The study design utilizes an ecologic study approach, which while such study designs leave a lot to be desired in epidemiology, the approach offers early insights and provides ideas and associations worthy of further explorations.  The authors are mindful of the limitations of such a study design and provide the most important caveats in the limitations area of the manuscript.

With regards to areas for improvement, there were a few moderate concerns. 

Moderate Concerns:

*Overall, the manuscript acknowledges in the limitations that there is the potential for the ecologic fallacy and reverse causation if interpreting the correlations observed as being causal. One area not clearly delineated is the sheer scope of the national aspect of these data used for the associations. Greater emphasis of local and regional health orders may be wise to describe or at least acknowledge as part of the limitations. Reference 13 had to do so in their evaluation. A lot of public health and disease prevention occurs at the local level, even in a pandemic. Greater inclusion or awareness of the patchwork of local health policies would be a presumably important for readers to acknowledge which could not be captured in this dataset at a granular level (and is somewhat related to the ecologic fallacy). 

Other moderate concerns are as follows:

Abstract: Line 14: “Which policies are associated with outcomes other than mortality rates remain unknown” is not entirely true. Maybe so in the study location for this ecologic study, but not worldwide. There have been a number of studies on policies and disease incidence and hospitalization (or severe disease).

Abstract: The abstract does not include any tangible results, only results interpretation/discussion. Are there any key/signature quantitative results that could be used?

Intro: Line 55-57: The assumption that the impact of the stringency index’s association with other paramaters beyond peak spread, etc. being unknown is a false assumption. These studies have been done albeit possibly differently than this manuscript. Greater caution or more thorough clarification than an overarching “is unknown” statement should be drafted to replace this statement. It is possible when the first manuscript draft was derived, these studies were not available, but they are available now and ought be considered, particularly the first two references of three shown below.

Wong, M.C., Huang, J., Teoh, J. and Wong, S.H., 2020. Evaluation on different non-pharmaceutical interventions during COVID-19 pandemic: An analysis of 139 countries. The Journal of Infection81(3), p.e70.

Ma, Y., Mishra, S.R., Han, X.K. and Zhu, D.S., 2021. The relationship between time to a high COVID-19 response level and timing of peak daily incidence: an analysis of governments’ Stringency Index from 148 countries. Infectious diseases of poverty10(1), pp.1-10.

Haldar, A. and Sethi, N., 2020. The effect of country-level factors and government intervention on the incidence of COVID-19. Asian Economics Letters1(2), p.17804.

Discussion: Line 267-269: Consider the following reference as it relates to contact tracing.  Specifically, contact tracing could only be effective with sufficient testing capacity as authors did mention (but consider citing a paper). Also, even high-income nations in some cases had disorganized/decentralized contact tracing strategies or inadequate personnel resources – which presumably were more challenging in low- or middle-income nations.

Clark, E., Chiao, E.Y. and Amirian, E.S., 2021. Why contact tracing efforts have failed to curb coronavirus disease 2019 (covid-19) transmission in much of the united states. Clinical Infectious Diseases72(9), pp.e415-e419. https://academic.oup.com/cid/article/72/9/e415/5881818?login=true

For some minor grammatical concerns: 

References: if the numbers in parentheses, like  (1), (2), (3), are references, then they should be in brackets. The guidelines for the journal are as follows: “In the text, reference numbers should be placed in square brackets [ ], and placed before the punctuation; for example [1], [1–3] or [1,3]  *See https://www.mdpi.com/journal/healthcare/instructions#preparation

Abstract Line 22: “Results were presented descriptively”  -- probably unnecessary statement for abstract

Abstract Line 26: ‘most’ should be ‘mostly’

Introduction: Line 39: the wave concept is not entirely appropriate in this statement, in the since that some regions are experiencing third waves at this point. The use of ‘wave’ terminology limits the prevention logic to just waves rather than disease spread in general from COVID.

Introduction: Line 40: The UK “has” implemented. The UK is treated as a singular term, and the verb “have implemented” is for plural.

Introduction: Line 50: the School of Government is a singular entity, therefore it “has developed” rather than “have developed”

Introduction: Line 59: insert “a” ---- using “a” profile approach

Introduction: Line 61: insert “a” --- which provides “a” better global assessment

Methods: Line 66: insert “the” --- the mean mortality rate during “the” rising phase of the curve

Methods 2.1. replace “:” with a period. “were analyzed.” Or replace “they were” with “including”.    …. were analyzed: including Algeria, Austria,….

Methods: It is very difficult to read the 2.2.1. section. It would be better to list in the statements what was measured. Then after listing all the items that were measured, then providing the definitions, rather than trying to list while also defining within the listing. This section needs more clear organization for enhancing readership. Possibly a table to list the measure name, then provide a definition?

Methods 2.2. (Line 135): “Countries” should be “Country income data” or “Each country’s income data”
